

# Emergent constraints for the climate system as effective parameters of bulk differential equations

Chris Huntingford[1], Peter M. Cox[2], Mark S. Williamson[2], Joseph J. Clarke[2], and Paul D.L. Ritchie[2]

[1]U.K. Centre for Ecology and Hydrology, Benson Lane, Wallingford, Oxfordshire, OX10 8BB, U.K.
[2]College of Engineering and Environmental Science, Laver Building, University of Exeter, North Park Road, Exeter, EX4 4QF, U.K.

*Correspondence to:* Chris Huntingford (chg@ceh.ac.uk)

**Abstract.** Planning for the impacts of climate change requires accurate projections by Earth System Models (ESMs). ESMs, as developed by many research centres, estimate changes to weather and climate as atmospheric Greenhouse Gases (GHGs) rise, and they inform the influential Intergovernmental Panel on Climate Change (IPCC) reports. ESMs are advancing the understanding of key climate system attributes. However, there remain substantial inter–ESM differences in their estimates of future

meteorological change, even for a common GHG trajectory, and such differences make adaptation planning difficult. Until recently, the primary approach to reducing projection uncertainty has been to place emphasis on simulations that best describe the contemporary climate. Yet a model that performs well for present–day atmospheric GHG levels may not necessarily be accurate for higher GHG levels and vice-versa.

A relatively new approach of Emergent Constraints (ECs) is gaining much attention as a technique to remove uncertainty

between climate models. This method involves searching for an inter–ESM link between a quantity that we can measure now and another of major importance for in describing future climate. Combining the contemporary measurement with this relationship refines the future projection. Identified ECs exist for thermal, hydrological and geochemical cycles of the climate system. As ECs grow in influence on climate policy, the method is under intense scrutiny, creating a requirement to understand them better. We hypothesise that as many Earth System components vary in both space and time, their behaviours often

satisfy large–scale Partial Differential Equations (PDEs). Such PDEs are valid at coarser scales than the equations coded in ESMs which capture finer high resolution gridbox–scale effects. We suggest that many ECs link to such an effective hidden PDE that is implicit in most or all ESMs. An EC may exist because its two quantities depend similarly on an ESM–specific internal bulk parameter in such a PDE, and with measurements constraining and revealing its (implicit) value. Alternatively, well–established process understanding coded at the ESM gridbox–scale, when aggregated, may generate a bulk parameter

with a common "emergent" value across all ESMs. This single parameter may link uncertainties in a contemporary climate driver to those of a climate–related property of interest, the EC constraining the latter by measurements of the former. We offer illustrative examples of these concepts with generic differential equations and their solutions, placed in a conceptual EC framework.



## 1 Introduction

Earth System Models (ESMs) form the basis of climate research and provide predictions of global environmental change due to burning fossil fuels. Projections by ESMs strongly inform the reports of the Intergovernmental Panel on Climate Change (e.g. IPCC, 2013, 2021) and influence climate policy. These models consist of solving, on numerical meshes, discretised differential

equations that describe the evolution of the atmosphere, oceans, land and cyrosphere and their interactions. In addition to physical processes, these models have evolved to emulate key global geochemical cycles. ESMs are typically forced with prescribed values of historical atmospheric greenhouse gas (GHG) concentrations, followed by a range of scenarios for their future levels (e.g., Meinshausen et al., 2011). This process estimates how the planetary climate system responds to altered atmospheric gas composition. Alternatively, an ESM can be forced with $CO_2$ emissions scenarios (e.g., Cox et al., 2000), if

the ESM has a full description of the global carbon cycle. A major achievement of the scientific community is the pooling of climate model projections from different research centres into common Coupled Model Intercomparison Project (CMIP) databases such as CMIP5 (Taylor et al., 2012) and CMIP6 (Eyring et al., 2016).

Almost all parts of the climate system vary in both space and time. Hence Partial Differential Equations (PDEs) are solved for evolving temporal variations on the spatial numerical mesh particular to any ESM. Many of these PDEs central to under-

standing the climate system are well–established, as described in standard textbooks on atmospheric and oceanic behaviours (e.g. Vallis, 2006). However, for the same future GHG scenario, analyses of the CMIP databases reveal significant inter–ESM differences between projections of even fundamental quantities such as the level of global warming (Lee et al., 2021). As standard equations are frequently solved in ESMs, a valid question is: "why are ESM projections often so different"?. The main possibly simplest answer is that some processes are still not fully understood and are therefore parameterised differ-

ently between ESMs. Components frequently noted in this category are the modelling of cloud–climate interactions (e.g. Bony et al., 2015), and how aerosols act in modulating global temperature rise (e.g. Bellouin et al., 2020). A secondary source of uncertainty is the dependence of process parameterisation on gridbox resolution. Larger individual gridboxes (i.e. a coarser numerical grid) often need effective parameterisation of sub–grid processes, and variation in this may cause inter–ESM differences. Numerical tests with extremely high resolution models allow the explicit representation of convection ('convection

permitting'; e.g. Clark et al., 2016) and verify its importance in describing local rainfall characteristics. While very high resolutions are achievable in weather forecast models, computational speed precludes their routine operation for ESMs designed to simulate century timescales.

Unfortunately, the considerable variation in model estimates of future climate change makes societal adaptation planning difficult. Such discrepancies can be used by some to discredit the overall notion of a human influence on climate. One possibility

to lower inter–ESM spread is to rank models by their ability to describe the contemporary climate and known recent changes. ESMs regarded as the most reliable at describing expected future change are those that perform best at simulating the recent past. However, this can be a subjective activity, depending on selected datasets for comparison and their geographical location. Furthermore, there is a risk of downrating a model that does not perform well for the present day yet accurately projects a future change of concern to society.



Recently a technique called "Emergent Constraints" (ECs) has gained prominence as a new method to reduce the spread between the projections by different ESMs. The EC method capitalises on discovered relationships between two quantities calculated by climate models when considering estimates of each from across many ESMs. One variable is an attribute of the climate system for the present-day or historical period, for which data also exists. The second variable, for which data is

unavailable, is often a feature of the evolving climate system and is informative for climate policy. For example, this second variable may be an internal sensitivity of the climate system that determines changes to mean meteorological conditions as GHGs rise. Alternatively, it can be the direct estimate of some feature of climate change (e.g. an aspect of near–surface meteorology) corresponding to specific future higher GHG levels. Measurement of the first quantity, in combination with the discovered inter-ESM link between the two variables (i.e. the EC), provides the constraint on the magnitude of the second

unknown variable.

The first application of the EC technique was to refine estimates of large-scale snow albedo feedbacks in a warming world (Hall, 2004). Since then, the EC method has lowered uncertainty in a substantial number of components of the Earth system (Hall et al., 2019), and including for fundamental climate quantities such as Equilibrium Climate Sensitivity (ECS) (e.g. Cox et al., 2018). Other researchers have provided EC–based estimates of both ECS and Transient Climate Response (TCR)

(Jimenez-de-la Cuesta and Mauritsen, 2019; Nijsse et al., 2020; Tokarska et al., 2020). Applications of ECs to physical parts of the Earth system have included cloud feedbacks (e.g. Klein and Hall, 2015), as well as components of global geochemical cycles. ECs on aspects of geochemical cycles include constraining the expected level of ocean acidification (Terhaar et al., 2020), marine primary productivity (Kwiatkowski et al., 2017) and soil carbon turnover (Varney et al., 2020). Notable is that for many discovered ECs, the variable for which measurements exist is often a statistic of a quantity fluctuating at shorter

timescales than longer–term climate–related variation the EC estimates. This use of high frequency variations highlights how ignoring system fluctuations may constitute disregarding valuable information about the climate system. The EC approach, therefore, offers an interesting comparison to the method of weighting ESMs by simply comparing their projections of present day trends against measurements, as the latter neglects variation about such trends.

With ECs becoming ubiquitous in climate research and with their potential to enable better decisions on GHG emissions that

avoid dangerous change, it is appropriate that the method be moved to a stronger scientific basis. Some recent papers review the EC method, highlighting its capability and listing a set of potential pitfalls. For instance, Williamson et al. (2021) identify a particularly broad range of discussion points related to ECs, all framed in their application to refining estimates of ECS. Further critiques of the EC method exist in the context of the terrestrial carbon cycle (Winkler et al., 2019), Arctic warming (Bracegirdle and Stephenson, 2012) and ECS (Caldwell et al., 2018) - all note potential issues that could result in incorrect

bounds on future estimates of change. Schlund et al. (2020) test the transferability of bounds derived for estimates of ECS (using different ECs) first created with models in the CMIP5 ensemble. These researchers find that the EC–based uncertainty bounds, when derived using the CMIP6 ensemble, are generally larger than when using the CMIP5 models. Suggested causes of this widening of uncertainty include the possibility that the spread of model projections informing an EC may be less in the CMIP6 ensemble, lowering the ability to find a tight constraint. A second possibility is that the CMIP5 models were overly

simplistic, and the CMIP6 models include better process representation, but this also introduces substantial new uncertainty



that weakens the ability of ECs to constrain ECS. Fasullo et al. (2015) provide an important discussion on whether it is expected that ECs hold across different generations of ESMs.

Yet despite recent scrutiny, there remains a basic, almost philosophical question: "What is an Emergent Constraint"?. While there are likely many perspectives on the answer to this question (see Nijsse and Dijkstra, 2018; Williamson et al., 2021, for
example), here we suggest that one way to interpret many ECs is that they derive bulk parameters associated with differential equations that are valid at large spatial scales. Such equations are implicit in ESMs (i.e. not coded explicitly) and instead "emerge" by aggregating the numerical finite difference schemes that are solved in ESMs at the finer gridbox spatial resolution. Here we hope to initiate a discussion of whether this is an appropriate way to describe the underpinning properties of many ECs. We consider simple illustrative examples using standard solutions to basic differential equations but in the novelty of
being placed in the context of the framework of the EC method.

## 2 Methods and Conceptual Examples

### 2.1 The Emergent Constraint Method

The core of any EC is the discovery of a robust link, across different ESMs, between a driving variable, say $X$, and another model calculated quantity, $Y$. Variable $X$ is a quantity for which contemporary measurements are available. Quantity $Y$ is a
climate–related statistic, metric or parameter often of importance for developing future adaptation or mitigation strategy, but for which data does not exist. It is the EC relationship between $X$ and $Y$, in tandem with the measurement of $X$, that constrains our understanding of $Y$. In general, it is considered preferable that ECs are found by process intuition that reveals related quantities, rather than direct inter–ESM "data mining". For instance, in the context of finding ECs to constrain understanding of the size of cloud feedbacks, Klein and Hall (2015) propose that each should be "accompanied by credible physical explanations". The
EC relationship between $X$ and $Y$ may take many forms, such as a nonlinear response, or potentially multidimensional with more than one $X$ component.

For illustration purposes, we imagine an EC that is a simple linear regression between two variables, and when indexing each ESM with $i$, is of the form:

$$Y_i = a_0 + a_1 X_i + \epsilon_i + \eta_i. \tag{1}$$

Here parameters $a_0$ and $a_1$ quantify the emergent constraint, and $\epsilon_i$ and $\eta_i$ are ESM–specific "noise" terms. We imagine that $\epsilon_i$ captures how far an individual ESM is from the fitted relationship of Eq. (1), and so any large absolute value corresponds to a model outlier. Quantity $\eta_i$ is a random variable, that describes natural climate variability for each model. Measurement $X^*$ utilises this relationship to predict the value of $Y$, named $Y^*$. Cox et al. (2018) provide the methodology to derive uncertainty bounds on the constrained value $Y^*$, which include being a function of both $\epsilon$ and the size of uncertainty bounds on data $X^*$.



## 2.2 Simple thermal "box" model with different heat capacities

Our working assumption is that ECs exist due to common inter–ESM deterministic processes, which we attempt to mirror with abstract but illustrative, simple models. As such, the noise quantities $\epsilon_i$ and $\eta_i$ are only reconsidered towards the end of our analysis, and then only visually. We start with an especially simple conceptual representation of an EC. We consider a set of single thermal box models indexed by $i$. This indexing may mirror the differentiation between ESMs in a collection of models, such as the CMIP6 ensemble (Eyring et al., 2016). Each model has a different heat capacity $c_{p_i}$ (J K$^{-1}$), in to which we assume there is a common and known forcing heat flux $H(t)$ (W). Long-term changes in this forcing are regarded as analogous to Representative Concentration Pathways (RCP) of future GHGs levels, often applied as an equal forcing across ESMs. As a single box, there is no spatial variation, so the model is regarded as having infinite diffusion. The equation for the box temperature $T(t)$ (K), where $t$ (year) is time, $c'_{p_i} = c_{p_i}/n_{y,s}$ (J K$^{-1}$ yr s$^{-1}$) and $n_{y,s}$ (s yr$^{-1}$) is number of seconds in a year, is:

$$c'_{p_i} \frac{dT}{dt} = H. \tag{2}$$

We first consider a known fluctuating heat flux, $H = b\cos(\omega t)$, for the contemporary period to force each model indexed by $i$. This could be interpreted as a form of known annual seasonal cycle (and therefore $\omega = 2\pi$), and this forcing results in a model–specific temperature, $T_i(t)$. In addition to the known common $H$ driver, observed are seasonal temperature features named $T^*$ (K). The simple solution to Eq. (2) with this periodic forcing is:

$$T_i(t) = C_0 + \frac{b}{c'_{p_i}\omega}\sin(\omega t). \tag{3}$$

Removal of background multi–year temperature allows the setting of arbitrary constant $C_0$ as $C_0 = 0$. Required is a simple statistic applicable to both modelled temperature projections and measurements, which could be the seasonal range, $\Delta T_S$ (K). Hence $\Delta T_{S_i} = \max(T_i) - \min(T_i)$, and so for each model and from Eq. (3),

$$\Delta T_{S_i} = \frac{2b}{c'_{p_i}\omega}. \tag{4}$$

Considered additionally is a longer–term forcing of our model, representing ongoing climate change. We describe this extra forcing as simply a fixed value of $H_0$ (W) for $t > 0$. Hence this gives a combined forcing of $H(t) = H_0 + b\cos(\omega t)$, and solving Eq. (2) for both drivers simultaneously gives:

$$T_i(t) = \frac{H_0 t}{c'_{p_i}} + \frac{b}{c'_{p_i}\omega}\sin(\omega t) \qquad t > 0. \tag{5}$$

A second temperature–based statistic we can make is the running mean from the solution of Eq (5) by averaging within individual years to remove seasonality. This running mean is the background change in $T$ and is analogous to long–term climate variation, such as global warming. Such averaging, denoted by an overline, is simply:

$$\overline{T_i(t)} = \frac{H_0 t}{c'_{p_i}}. \tag{6}$$





A possible EC is now revealed. The issue of future concern might be the rate of change of mean temperature $T_i$. Plotting for the simple model an "$x$" axis of $\Delta T_{S_i}$ (from Eq. (4)) and a "$y$" axis of $\mathrm{d}\overline{T_i(t)}/\mathrm{d}t = H_0/c'_{p_i}$ (from Eq. (6)) would yield a diagram where both quantities increase, linearly, in $1/c'_{p_i}$. The EC is, therefore, a relation between seasonal temperature variation and long–term warming that holds across all $c'_{p_i}$ values. Knowledge of the actual $x$ axis variable, which here would be the known

observed seasonal amplitude, $\Delta T_S^*$, constrains the bounds of the uncertainty of the $y$ axis quantity. We present these ideas schematically in Fig. 1, and show the uncertainty, $\epsilon_i + \eta_i$, as just random distances by individual models (black dots) away from the EC regression line.

In the analysis presented above, the parameters related to forcings, i.e. $b$ and $H_0$, are assumed to be invariant between models. Hence the measurement in tandem with the EC is designed to lower uncertainty on the model–specific value of bulk

parameter $c'_{p_i}$. However, an alternative possibility is an emergent constraint where there is instead uncertainty in the magnitude of the forcings between models (for instance, a range of representations, between ESMs, of the translation of atmospheric aerosol levels to their cooling effect). Now the forcing parameters are indexed as $b_i$ and $H_{0_i}$. Subject to $b_i/H_{0_i}$ being invariant between models, an EC of identical form to that of Fig. 1 could exist where instead $c'_p$ has a single numerical value, common to all models. In this case, the emergent constraint represents the discovery that there is a single model–independent internal bulk

parameter (i.e. $c'_p$), while the data point constrains uncertainty in the forcing element $b_i$. With the forcing uncertainties common for both short– and long–term drivers, the data therefore also constrains $H_{0_i}$ and thus the background warming $\overline{\mathrm{d}T_i/\mathrm{d}t}$.

## 2.3 Thermal model with spatial variation

We extend the basic box model of Section 2.1 with a further illustrative example that introduces spatial variability via directional coordinate $x$ (m) and retain temperature as our notional state variable. Now we consider the system to evolve on a semi–infinite

domain $0 \leq x \leq \infty$, and with the heat forcing boundary condition at $x = 0$. This framework may depict, for instance, heat absorption by the oceans and where information on future trends in surface temperature is required. Specifically, we solve for $T_i(x,t)$ as satisfying a diffusion equation:

$$c'_{p_i} \frac{\partial T_i}{\partial t} = \kappa_i \frac{\partial^2 T_i}{\partial x^2} \qquad 0 \leq x \leq \infty. \tag{7}$$

Here $c'_{p_i}$ (J K$^{-1}$ m$^{-3}$ yr s$^{-1}$) remains a form of heat capacity, while $\kappa_i$ (W m$^{-1}$ K$^{-1}$) is a conductivity or mixing parameter,

and both parameters may be model specific, as indexed by $i$. We again start by prescribing a seasonal boundary condition, at $x = 0$, given by:

$$\kappa_i \left. \frac{\partial T_i}{\partial x} \right|_{x=0} = -H = -b\cos(\omega t). \tag{8}$$

The solution to governing Eq. (7) with the boundary condition of Eq. (8), assuming no non–seasonal transient terms and that $T_i$ is bounded as $x \to \infty$, is:

$$T_i(x,t) = \frac{be^{-\left(x\sqrt{\frac{c'_{p_i}\omega}{2\kappa_i}}\right)}}{\sqrt{c'_{p_i}\kappa_i\omega}} \cos\left[-\omega t + \frac{\pi}{4} + x\sqrt{\frac{c'_{p_i}\omega}{2\kappa_i}}\right] + C_0. \tag{9}$$





**(a)**

Forcing : $\qquad H(t) = \qquad H_0 \qquad + \, b\cos(\omega t)$

Model : $\qquad c_i \dfrac{\mathrm{d}T}{\mathrm{d}t} = H$

Response : $\qquad T_i(t) = \qquad \dfrac{H_0 t}{c'_i} \qquad + \dfrac{b}{c'_i \omega} \sin(\omega t)$

**(b)**

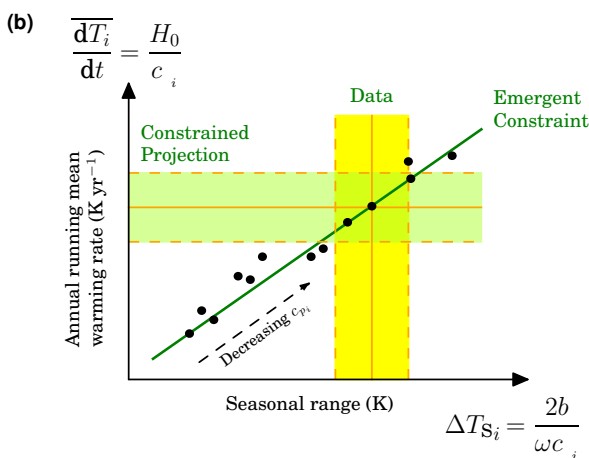

Figure 1. **Schematic representation of a simple emergent constraint**. Panel (a) (top row) shows the combined equation for long–term and seasonal forcing (so with $\omega = 2\pi \ \mathrm{yr}^{-1}$) driving the thermal box model given by Eq. (2) (middle row), and the related response to both forcings, which combine additively to give Eq. (5) (bottom row). Panel (b) illustrates a related emergent constraint, based on the response Eq. (5), as also shown in panel (a). This response contains a seasonal ($x$ axis) and long–term ($y$ axis, with seasonality ignored), and the EC links the two. The EC allows the observation of seasonal fluctuations to constrain the long–term rate of change of state variable, $T$. Each model (black dots, indexed by $i$) has a different implicit value for $c'_p$ i.e. $c'_{p_i}$. The EC is assumed to not be exact, with noise causing variation around the regression line (the $\epsilon_i$ and $\eta_i$ terms of Eq. (1)). The vertical yellow band represents uncertainty in the measurement, $\Delta T_S^*$. The constrained projection of the long–term warming rate (based on the EC, the value of $\Delta T_S^*$ and its uncertainty) is given by the green horizontal band.

Hence the value of $T_i$ at $x = 0$, with additive constant set to $C_0 = 0$, is given by:

$$T_i(0,t) = \frac{b\cos(-\omega t + \pi/4)}{\sqrt{c'_{p_i}\kappa_i \omega}}. \tag{10}$$



From Eq. (10), the temperature seasonal cycle at $x = 0$ corresponds to a range of:

$$\Delta T_{S_i} = \max(T_i(0,t)) - \min(T_i(0,t)) = \frac{2b}{\sqrt{c'_{p_i} \kappa_i \omega}}. \tag{11}$$

and for which we consider there is similarly a corresponding observable value, $\Delta T_S^*$.

In further analogy to our example with the box model example, we consider an additional long–term heat flux, $H_0$ at $x = 0$,

starting at time $t = 0$. That is, a boundary condition of:

$$\kappa_i \left. \frac{\partial T_i}{\partial x} \right|_{x=0} = -H_0 \qquad t > 0. \tag{12}$$

and this has a solution of:

$$T_i(x,t) = \frac{2H_0}{\kappa_i} \left[ -\frac{x}{2} \operatorname{erfc}\left( \frac{x}{2} \sqrt{\frac{c'_{p_i}}{\kappa_i t}} \right) + \sqrt{\frac{\kappa_i t}{\pi c'_{p_i}}} e^{-\frac{c'_{p_i} x^2}{4\kappa_i t}} \right] \qquad t > 0, \, x > 0. \tag{13}$$

The solution to Eq. (13) at $x = 0$ corresponds to:

$$T_i(0,t) = 2H_0 \sqrt{\frac{t}{c'_{p_i} \kappa_i \pi}} \qquad t > 0. \tag{14}$$

As our governing Eq. (7) is linear, the seasonal and long–term solutions (Eqs. (9) and (13) respectively) may be simply added. Hence a combined heat flux in to the system of $b\cos(\omega t) + H_0$ at $x = 0$ generates a surface temperature $T_i(0,t)$, for $t > 0$, given by the addition of Eqs. (10) and (14). The inclusion of spatial variation, via $x$, causes a long-term transient effect where although the long–term average heat flux is constant, the surface temperature given by Eq. (14) has a $\sqrt{t}$ response. This solution

compares to a linear long–term temperature response for our single box model example in Eqs. (5) and (6).

For our example with spatial variation, a possible emergent constraint could constitute an $x$ axis of $\Delta T_{S_i}$ (Eq. 11) and a $y$ axis of $\overline{dT_i(0,t)}/dt \times \sqrt{t} = H_0/\sqrt{c'_{p_i} \kappa_i \pi}$ (differentiation of Eq. (14) with respect to time, in tandem with averaging out the seasonal variations of Eq. (10)). Using these variables, both the $x$ and $y$ axes are linear in $1/\sqrt{c'_{p_i} \kappa_i}$ for the different indices $i$. We present this EC schematically in Fig. 2. In conjunction with this EC, knowledge of seasonal temperature variation reveals

and so constrains the long–term warming rate. In this example the data point constrains, implicitly, the value of $c'_{p_i} \kappa_i$. If $c'_{p_i}$ is well known and fairly invariant between ESMs, then the data point is constraining the implicit value of $\kappa_i$, or vice versa where the constraint is on $c'_{p_i}$

In a strong similarity to the discussion of uncertainty in the forcing boundary conditions of the box model and their potential constraint, the same possibility exists for our example with spatial variability. In the event that both effective parameters $c_p$

and $\kappa$ show little or no variation between ESMs, but there is uncertainty in $b$ of Eq. (8) and $H_0$ of Eq. (12) (and with identical uncertainties, so again, $b/H_0$ is invariant inter–ESMs), then the EC combined with data for $\Delta T_S$ acts to remove that uncertainty. Such removal of uncertainty between ESMs associated with forcing, via the EC and measurement of $\Delta T_S$, again constrains longer–term warming levels in this illustrative example.




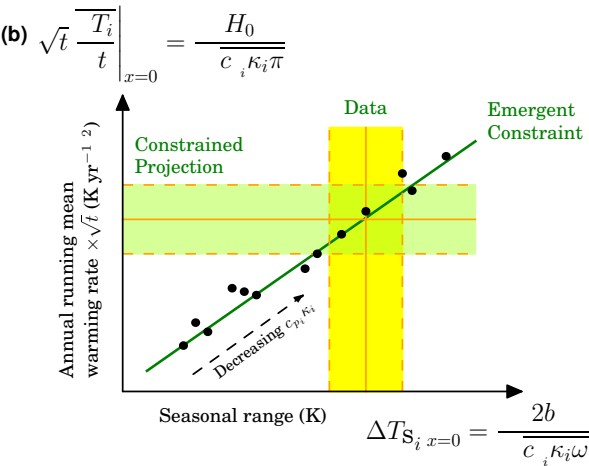

**Figure 2. Schematic representation of an emergent constraint with a spatial component**. The spatial dimension is defined by $x$. Panel (a) (top row) shows the combined equation for long–term and season forcing at $x = 0$, driving the diffusive model given by Eq. (7) (middle row), and the related response at $x = 0$ and $t > 0$ given by Eqs. (10) and (14) (bottom row). The seasonal forcing (so with $\omega = 2\pi$ yr$^{-1}$) is given by Eq. (8) and the long–term forcing to the thermal model given by Eq. (12). These two forcings generate a response in $T$ at $x = 0$ given by Eqs. (10) and (14) respectively, that combine additively and as shown. Panel (b) illustrates the related emergent constraint, based on the response $T_i(0, t)$ shown in panel (a). This response contains a seasonal ($x$-axis) and long–term ($y$ axis, with seasonality ignored) part, and the EC links the two. The EC allows the observation of seasonal fluctuations to constrain the long–term rate of change. Each model (black dots, indexed by $i$) has a different implicit value for $c'_{p_i} \times \kappa_i$. As for the example of Fig. 1, the EC is again assumed to not be exact, with noise causing variation around the regression line. The vertical yellow band represents uncertainty in the measurement of $\Delta T_S$. The constrained projection of the long–term warming rate (multiplied by $\sqrt{t}$, and based on the EC, the value of $\Delta T_S$ and its uncertainty), is given by the green horizontal band.



## 3    Discussion and Conclusions

How climate will change due to the ongoing burning of fossil fuels remains one of the most high–profile questions asked of the scientific community. ESMs are central to such research activity, and their primary objective is to accurately predict climate change for different potential future GHG levels. However, substantial differences can exist between ESM projections, even for the same future scenario of atmospheric GHG changes, so dependable methods are required to reduce the spread in simulations. Emergent constraints are discovered linkages, inter–ESM, between a quantity that is presently measured and a second important climate attribute associated with future changes, and where data on the former constrains our assessment of the value of the latter. With constant pressure to provide policymakers with refined estimates of future climate change, and against the backdrop of considerable variation between ESMs, ECs have attracted substantial application to a plethora of components of the Earth system. The rapid rise in EC discoveries and their near–ubiquitous use to constrain uncertainty enables a way to extract additional information from available ESMs that have required huge expenditure to build and operate. However, with such a high prominence of ECs as a method to lower uncertainty, it is timely to investigate the assumptions that underline them and any potential pitfalls (e.g. Williamson et al., 2021). Here we try to start an additional but related route of investigation. We suggest a potential explanation of many ECs is that their basis relates to solving large–scale equations that are both implicit in ESMs and have common features between models.

We develop the hypothesis that many identified ECs relate to undiscovered differential equations that describe the Earth System at large geographical scales. Such equations are not coded explicitly in ESMs, but instead "emerge" as the aggregation of the finer resolution behaviour of the climate system. Such finer resolution features are calculated in ESMs as the solution of differential equations solved on the numerical mesh of each model and capture environmental processes that are often understood well. Such understanding introduces similarities between models, which remain present in any spatial aggregation. The role of ECs is to enable the discovery of the implicit value of parameters associated with such large–scale equations, where uncertainty remains. Such bulk parameters affect both a quantity of interest linked to predicting future climate and a contemporary attribute of the Earth system. The contemporary quantity is measurable and, in tandem with the EC, constrains the parameterisation and thus understanding of the quantity associated with the future. In many instances of discovered emergent constraints, the present–day component is of a higher frequency fluctuation (e.g. seasonal), with the EC then projecting a climate attribute of relevance to decadal or century timescales.

We have presented two illustrative examples of solving standard differential equations but placed them in a structure as if they reveal an emergent constraint. We imagine the equations to be underlying large–scale bulk equations, solved implicitly in multiple ESMs, as outlined above. There are many examples of equations that represent the aggregated behaviour of fine–scale systems. For example, the bulk properties of an ideal gas, temperature and pressure, are related through the ideal gas law. However, these bulk properties can also be understood as the aggregated behaviour of the molecules (their mean velocity, mass and number density) that make up the gas. Formally these relations can be made through kinetic theory (Pitaevskii and Lifshitz, 1981). There are also examples of linear bulk dynamics emerging from nonlinear fine scale dynamics as well as the converse





- effective nonlinear bulk behaviour from linear microscopic dynamics e.g. the phase transition in the two dimensional Ising model (McCoy, 1973).

Our first case is a simple box model for which we wish to derive a thermal capacity term, $c_p'$, and the second has a single spatial variation and represents a search for a multiplicative combination of capacity and diffusion, $c_p'\kappa$. A discovered EC

between models, combined with measurements, would reveal the actual real world value of $c_p'$ or $c_p'\kappa$. Large values of $\epsilon_i$ are for models that are outliers to the EC. In the context of our abstract examples, outliers have different values of effective parameters $c_p'$ or $c_p'\kappa$ dependent on whether considering shorter seasonal timescales or longer periods, and implies these models to have substantially different process representation compared to most other ESMs. We also suggest an additional EC possibility where effective parameters emerge as invariant between ESMs and instead there is uncertainty in forcings (here, $b$ and $H_0$, although

the uncertainty is identical between the two parameters). Our conceptual model determines internal system properties, i.e. parameters, which for the spatial example are constrained based on behaviours at the edges of the domain. We note the basic theorems of vector calculus (e.g. Stokes' theorem) that relate integrated internal system features to conditions along domain edges.

A broad set of possibilities may link to our suggestion that the underlying principle of many ECs is the existence of equations

valid at the large scale. For instance, additional to our example of diffusion, PDEs with spatial dimensions can also simulate advection, and in terms of climate modelling this may correspond to teleconnections. To constrain the strength of future teleconnections is likely to need a present day measurement of wind fluxes, or measurements of a quantity of interest in two locations. Modelling many components of the Earth system requires coupled differential equations to link different physical quantities, capture changes of state, or where geochemical cycles link tightly to climate variation. An example of an EC cap-

turing features of a coupled system is that of Cox et al. (2013). In that analysis, data on present–day simultaneous fluctuations in atmospheric $CO_2$ and annual temperature anomalies reveals the fate of future South American carbon stores under global warming and the related risk of the iconic possibility of Amazon forest "die–back". In some cases, the EC $x$ axis, for which measurements exist, is a combination of high-frequency drivers and response, and for the same variable. As an example of such a more refined and complex contemporary statistic, Cox et al. (2018) estimate equilibrium climate sensitivity with a statistic $\Psi$

that is a combination of the standard deviation and autocorrelation of current global temperature fluctuations. Arguably, the $\Psi$ statistic merges a system driver (standard deviation) and a response (autocorrelation).

In summary, the analysis of ensembles of ESMs, as built by different research centres, has revealed multiple emergent constraints for all parts of the Earth system (Hall et al., 2019). Discovered ECs have reduced uncertainty bounds for features of the climate system that directly affect society and are therefore of particular interest to policymakers. With the placement of

much emphasis on the EC method to lower uncertainty, there is a growing requirement to understand its underlying assumptions better. Timely research is emerging that critically assesses the method (e.g. Williamson et al., 2021). We wish to add to the discussion by suggesting that many ECs represent the discovery of parameters associated with large–scale implicit equations that describe features of the Earth system. Such equations emerge from the aggregation of more local effects simulated on the gridpoints of the numerical meshes of individual ESMs. We do not propose this as a universal theory of ECs, as some

may function well for other reasons. However, with the general view that physical intuition provides a better route to EC





discovery than, say, data mining, our suggestion is therefore analogous to that standpoint. Hence we consider most ECs to correspond to underlying processes and related mathematical representation. Such bulk process discovery helps counter a view that ESMs are so complex that they can never be amenable to interpretation via standard applied mathematics techniques (a concern raised by Huntingford, 2017). Such methods include equation scaling ("nondimensionalisation") to find the dominant

underlying forms, although we speculate that EC discovery may instead identify key large–scale processes. Further hinting at the need to confirm underlying processes is the analysis of Qu et al. (2018), who consider the statistical linkages between four different ECs proposed for ECS and suggest that the discovered commonalities are because each is constraining, implicitly, shortwave radiation cloud feedbacks. We present two simple illustrative examples of differential equations, their solutions, and their potential interpretation as ECs. Despite differential equations representing a range of processes, mathematics can

often characterise them in discrete ways (for instance, every PDE being either diffusive, elliptic or hyperbolic). We conjecture that there are one–to–one mappings between ECs and equation forms and their identification could also incentivise revisiting aspects of climate change science from an applied mathematics standpoint. Although our examples are synthetic, we hope the concepts we present may support the placement of ECs on a stronger theoretical footing by, where applicable, revealing underlying bulk equations that fit with process intuition. We note Brient (2020) argue that when multiple ECs exist to predict

the same quantity, each should be weighted by the level of physical understanding they offer to elucidate the relationship. It remains important to understand ECs as they offer an elegant and nearly unique potential capability to lower the continuing uncertainty between ESM projections.

## 4   Code availability

The computer scripts leading to checking of the analysis solutions (with the sympy python module), and any of the diagrams

(with the matplotlib python module) are available on request to Chris Huntingford (chg@ceh.ac.uk)

*Author contributions.* C.H. devised the conceptual model framework, the format of the paper and undertook finding the exact solutions to the representative equations. All authors contributed to the writing of the manuscript, and placing the analysis in the context of existing emergent constraints.

*Competing interests.* The authors confirm they have no competing interests.

*Acknowledgements.* C.H. acknowledges the Natural Environment Research Council National Capability award to the U.K. Centre for Ecology and Hydrology (UK-SCAPE, NE/R016429/1). C.H., P.M.C. and M.S.W. acknowledge the European Research council (ERC) ECCLES project, Grant Agreement Number 742472.




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
