# Peer review of "Emergent constraints for the climate system as effective parameters of bulk differential equations"

_Earth System Dynamics, 2022_

## Author Comment (AC1)

Please find our proposed changes to manuscript "Emergent constraints for the climate system as effective parameters of bulk differential equations" in response to this review. Our replies are in blue font and indented.

Review of Huntingford et al 2023:

On one hand, this paper is clear, well-written, and its PDE examples are simple, relevant, and pleasant to work through. On the other hand, I didn't really learn anything from reading this. For example, I can't imagine anyone understanding Cox et al (2018) without having a deep understanding of the notion that the emergent equations governing temperature changes on various timescales are linked via heat capacity. This left me wondering whether the paper is worth publishing. Ultimately, I think the answer is yes because if someone didn't intuitively understand that emergent constraints occur due to links between underlying governing equations, this paper would do a nice job of introducing them to the concept. I doubt the paper will be cited much, but that doesn't mean it shouldn't be published.

> First, we thank the reviewer for their time assessing our manuscript. We are fully aware that this manuscript will only appeal to a subset of those using Emergent Constraints (ECs) to refine the understanding of climate system components. However, we regard that group as important, representing many with a mathematic interest asking: "What underpins the EC method?". With such a high profile of ECs to constrain uncertainty, we hope of paper will trigger new lines of questioning and understanding of the technique.
>
> Our proposed replies are below, indented and in blue.

Minor comments (Note my convention is P2 L1 = Page 2, Line 1):

1. P2 L1: observationalists would disagree that ESMs form the basis of climate research. I tend to say they're a pillar of climate research.

   > We will adjust the manuscript to use the word pillar.

2. P2 L6: It's not accurate to say that ESMs are typically forced with historical and scenario GHGs. A lot of time is spent on PI control, abrupt4xCO2, 1%CO2, etc. Minor rewording is needed.

   > We will note the extensive and often long PI control simulations in the CMIP databases, as well as factorial experiments such as abrupt4xCO2.

3. P2 L19: "main possibly simplest answer" is awkward grammar

   > We will split the sentence and reword as "…*why are ESMs different….. The simplest answer is….*"

4. P3 L30 – P4 L2: The first sentence here isn't very clear. I think you are saying that Schlund and others found that ECs based on CMIP5 were generally worse when applied to CMIP6. As written, it sounds like any EC, including ECs developed from CMIP6 data, would have wider bounds. I also found your wording a bit confusing because wider bounds could come from worse correlations between EC predictor and predictand OR from larger spread in the observations used to constrain. I guess the problem must be the former, but it takes the reader some unnecessary thought to get to that conclusion. Following on this, I think the obvious explanation for larger spread in CMIP6 is that the ECs from CMIP5 were overtrained: they are capturing noise rather than real EC signal. I'm confused how this possibility isn't even in your proposed reasons at all.

We agree that this text was not as well phrased as it should have been. In our revised manuscript, we will therefore replace lines P3 L30 to P4 L3, with:

*"Schlund et al. (2020) tested the robustness of proposed emergent constraints by out-of-sample testing on a different model ensemble. These researchers found that emergent constraints on ECS, which were developed using the CMIP5 ensemble, do not provide useful constraints on ECS in the CMIP6 models. These ECs, therefore, fail to be "confirmed" (Hall et al., 2019). Recognising the danger of arriving at spurious emergent constraints based on the results of relatively small model ensembles (Caldwell et al., 2010), Williamson et al. (2021) have set the challenge of deriving more robust theory-based emergent constraints. To inform attempts to meet that challenge, here we address the basic, almost philosophical question: "What is an emergent constraint?""*

5. P5 L15-16: you introduce T* here but don't use it again except P8 L3. I suggest deleting both T* references. In particular, the wording of the first intro to T* was very confusing (and I think, unnecessary).

   Our intention is that "*" represents measurements. We will reword this sentence and make the notation clear. Rather than adjust this notation, we will instead annotate the word "Data" above the vertical uncertainty bars in Figure 1 and Figure 2 to be "Data, $\Delta T_S$*".

6. P5 L26-28: When you say "running mean", I immediately wonder what the averaging period is. I think it would be better to call this statistic the "annual average". Relatedly, the running mean itself isn't a measure of climate change. The time derivative of the running mean is your proxy for climate change. But of course, the annual average isn't special in this regard – the long-term average of the time derivative of the instantaneous T(t) equation would give the same answer because the derivative is a linear operator.

   We will reword this as: *"A second temperature-based statistic we can consider are changes in the annual averages. The time derivative of annual averages is a proxy for global warming. Annual averaging, denoted by an overline, is…"*

7. P6 paragraph starting L8 and P8 paragraph starting L23: I think this discussion can be improved. I think the big point you're trying to make is that while the fact that there exists a predictive relationship between the observable and the future quantity of interest allows you to predict ECS, the slope of that relationship provides interesting information about the physical equations that underpin that relationship. I think you are further pointing out that even though there may be uncertain terms in the equation governing the current-climate variable and in the equation governing future change, those uncertain terms sometimes cancel out when the quantity you're actually interested in is the ratio between predictor and predictand. As it stands, I don't think it is interesting that uncertainty in either of 2 parameters would give rise to the intermodel spread needed to compute an emergent constraint. I also don't think you adequately explained why $b_i/H_{0i}$ would be constant across models.

   We agree that in a standard research paper on ECs, then finding a relationship between the observable and future quantity of interest is sufficient. Here, we try to provide a potential mathematical explanation for why such regressions emerge. We hope our suggestions will open new ways to interpret and understand ECs, which matters given their widespread use in climate science. Based on this reviewer's comment, we will add a sentence to this effect in the Discussion.

   We are keen on an example where the aggregation of internal behaviours may involve a single effective parameter that is invariant, inter-ESM. In that case, uncertainty may be in the forcing. To maintain a single degree-of-freedom, we consider this uncertainty to be in the

forcing and identically between seasonal and long-term forcing (i.e. this implies $b_i/H_{0i}$ fixed between ESMs). We accept that this appears slightly contrived, but it does allow us to illustrate a key point in our conceptual structure. We will, though, enhance the sentence to make this much clearer at the point where uncertainty in forcing is introduced.

8. This is a minor point, but the seasonal cycle in eq 4 and eq 11 won't be exactly equal to the observed seasonal cycle in a warming planet ($H_0>0$) because the planet will have warmed a bit in the 6 months between winter and summer.

    We will put in a sentence to this effect.

9. P8 L17: defining your current-climate metric as d/dt(annual-ave T(x=0)) makes sense, but multiplying it by sqrt(t) seems contrived. If you have such an exact understanding of the underlying equations, you'd probably already know what $H_0$ was, so a regression would be unnecessary!

    Process knowledge of any heat conduction (i.e. "parabolic") PDE will indeed suggest that sqrt(t) behaviour is often present. Ideally, the "y"-axis of any emergent constraint is independent of time (or scenario), as shown. But this does raise an interesting yet rarely mentioned point, that in some instances, an EC may be robust but its position on an 'x-y' graph depends on a future time or GHG level. We will add an additional sentence that makes this point, noting that should the 'y'-axis be simply dT/dt, then the EC would move downwards by 1/sqrt(t).

10. P11 L34-P12 L1: I can't imagine how a real emergent constraint wouldn't have a physical underpinning that can be expressed as an equation. We may not know what that equation is, but if there truly isn't an underlying equation behind an empirical relationship, how could that relationship possibly be real?

    The original paper version contains the sentence (P11, L34): "We do not propose this as a universal theory of ECS, as some may function for other reasons". Based on this reviewer's comment, we will remove this sentence. The manuscript will then lead directly into the next sentence that discusses the importance of an underlying physical process. We will ensure that sentence is written more strongly, noting we should always be pursuing explanatory physical mechanisms in EC research.

I don't think the emergent relationships that distill into an EC are necessarily (or even typically) PDEs. In your examples, the fine scale governing equations are PDEs, but the equations you derive for seasonal cycle and warming tendency are not. Similarly, concepts like "if you don't have much cloud in the current climate, then you don't have much cloud to lose in the future" are fundamentally connecting model state to model change. This doesn't invalidate any of this work, but a reframing of the title and some rewording in the abs

    OK, some large-scale implicit equations may be ODEs (as per our first example) or, indeed, simply algebraic. In the other direction, the overall system may still be relatively simple but involves coupled equations. We will modify the Abstract to remove mention of "PDEs" and replace it simply with "*equations*". An extra sentence will be added to the Discussion to note that the equations may take many formats.

---

## Author Comment (AC2)

The paper "Emergent constraints for the climate system as effective parameters of bulk differential equations" by Chris Huntingford et al. provides a formal description of emergent constraints as parameters of large-scale partial differential equations (PDEs). In contrast to small-scale PDEs explicitly coded into Earth system models (ESMs), these large-scale PDEs are not directly included in the models, but emerge across ESMs when aggregated across larger scales. Huntingford et al. provide two example PDEs derived from simple thermal models. By assuming different bulk parameters (e.g., heat capacities) for the different ESMs, they show that these PDEs can be used to derive emergent relationships between short-term and long-term responses of the system, which ultimately can be used as emergent constraints with appropriate measurements of the real Earth system.

> First, we thank the reviewer for their time assessing our manuscript.
>
> We appreciate the reviewer's summary above. As noted, our view of many ECs is that the "emergent" property is the discovery of large-scale differential equations coded implicitly in ESMs (via aggregation of explicit coding at finer scales). To be clear, we will amend the Abstract sentence at line 17 to read: "*We suggest that many ECs link to effective hidden PDEs implicit in ESMs and which aggregate small-scale features*"

**General Comments**

This paper reads well and provides an interesting approach that allows the derivation of emergent constraints from bulk PDEs. I agree with the authors that an emergent constraint discovery method based on physical reasoning and mathematical models is much more desirable than data mining, and will eventually lead to more credible and robust emergent constraints. However, I have some concerns about the relevance of this study regarding "real" emergent constraints.

> We are grateful that the reviewer thinks our paper reads well and is an interesting approach. We take full account of their concerns listed below, responding in full. Our replies are in blue font and indented.
>
> Concerning some of the more technical points, please note that there was an issue with the diagram .pdfs and the ESD online converter. The correct diagrams are presented below, and if our manuscript is accepted, we will work carefully with ESD to make sure they are reproduced as expected.

Currently, a large part of the argumentation of the paper is based on two very simple PDEs. Especially in the context of a changing climate (which is a necessary condition here), I think the equations are too simplified. Since the PDEs are missing a "loss" term, a constant forcing will lead to an infinitely rising temperature, which is not realistic. For example, what happens if you add linear loss terms (linear feedback) $-\lambda*T$ to your PDEs (e.g., so that your eq. (2) is similar to eq. (1) of Cox et al. 2018)? Could you still derive the emergent relationships from these new equations? I can imagine that there are certain conditions (e.g., small times, small $\lambda$, large forcings, …) under which your original equations are good approximations, but it would be good to guide the reader in detail through this process. Additionally, it would be very helpful if you can provide more details on these emerging bulk equations themselves and why they should be present in an ensemble of ESMs. Do you have any recommendations how to find such PDEs? An example with a real emergent constraint would also be incredibly helpful. All this will ultimately help the reader to gain more trust in your framework.

> The reviewer asks some fascinating questions here but answering these is cutting-edge research that is beyond the scope of this initial short perspective paper. Instead, we will add text to acknowledge the nature of the challenge that the reviewer poses. Specifically, we will add additional lines of text as:

*"PDEs emerge commonly where state variables are globally conserved (i.e. for state variables closely related to energy and momentum). To aid transparency, we have also assumed underlying PDEs that are simple by design. Making these underlying models more relevant to the Earth's climate is an outstanding challenge. For example, in addition to horizontal heat transport, our planet emits longwave radiation to the wider universe. Such radiation provides the restoring force, λ, that ultimately stabilises the near-surface temperature. Including such a restoring force in our simple PDE models is one possible extension of our analysis, although, in tandem with an unknown heat capacity, $c_p$, this would potentially generate a two-dimensional EC. In practice, fitting a two-dimensional EC may be challenging given the relatively small number of data points (i.e. individual ESMs). Furthermore, analytical solutions may exist that allow for a time-varying value of H that approximates known historical climatic forcing"*.

Finally, two technical comments: first, it would be very helpful if you could use continuous line numbers (and not start with "1" on every page) and also add line numbers to figure captions. Second, please consider depositing your code in a publicly accessible repository (e.g., Zenodo) to make your analysis more transparent and reproducible for other researchers.

Unfortunately, I think the ESD template for submission causes this form of line numbering. Final ESD papers have no line numbers. We are very happy to upload our code to a standard scientific repository.

**Specific Comments**

1. P.2, l.30: Maybe add a reference here? E.g., Knutti et al. (2017), https://doi.org/10.1002/2016GL072012

   Thank you. We will add this reference.

2. P.3, l.4: It would be more precise to refer to "observational" data here (alternatively "observation-based").

   We will make this suggested wording alternation.

3. P.3, l.12: A better reference for this might be Hall & Qu (2006), https://doi.org/10.1029/2005GL025127. You might also want to cite Allen & Ingram (2002), https://doi.org/10.1038/nature01092 here.

   We will add these two references and associated additional wording around their citation.

4. P.4, l.1-2: It might be helpful for the reader to add the key conclusion(s) of the discussion of Fasullo et al. (2015) you mention here.

   Yes, the Fasullo paper is important and interesting, and we will cite further its key findings.

5. P.4, l.29: I guess technically it's a function of the total noise, so ε **and** η, not only ε.

   Correct. That should read ε+η.

6. P.5, l.18: Required for what?

   We will rewrite this as: *"ECs require a quantity that is both modelled for the contemporary period and is available as a measurement, such as the seasonal range, $\Delta T_S$."*

7. P.6, l.15: It's not only the data points (I guess by "data points" you are referring to the (x, y) tuples you get from the models?), but also the measurements that constrains the forcing element b.

Please see our response below, which concerns the same sentence.

8. P.6, l.15-16: I think this sentence is not clear enough: "With the forcing uncertainties common for both short– and long–term drivers". You need to explicitly assume that $b_i/H_{0i}$=const across models; you should mention that.

We will rewrite this sentence (and split it into two), as well as adjust the following sentence to: "*In this case, the emergent constraint represents the discovery that there is a single ESM-independent internal bulk parameter (i.e. $c_p$'). Measurements then provide the constraint to remove uncertainty in the forcing element $b_i$. With the forcing uncertainties common for both short-and long-term drivers (i.e. the assumption that $b_i/H_{0i}$ is constant), the measurements implicitly constrain $H_{0i}$, and thus the background warming, $dT/dt$.*"

9. P.6, eq. (8): You might want to refer to Fourier's law here.

Yes, we will do that.

10. P.8, l.17: Why don't you simply divide T(0, t) by sqrt(t) to get a y that is not dependent on t?

Yes, we do exactly that scaling as the '$y$'-axis of the EC (please see Figure 2). Please note that our other reviewer encouraged the opposite, of not normalising by sqrt(t). We hope the current framework of keeping in the sqrt(t) in the text but normalising in the EC diagram (so making the EC time-independent) is a satisfactory presentation.

11. P.12, l.10: I think this classification only applies to linear second-order PDEs, not to every PDE.

Yes, we will make that point clear.

12. P.12, l.10-12: Can you elaborate what you exactly mean by these "one-to-one mappings" and why this should be the case? This is not clear to me.

We will rewrite this in more straightforward language. We are trying to say that if our paper encourages discovering EC underpinning beyond physical intuition, instead with a more rigorous mapping to differential equations, then such equations can be characterised by standard mathematic terminology.

**Technical Corrections**

1. P.3, l.19-20: The second part of this sentence is hard to understand, please rephrase.

We will write this sentence more simply. We will point out that many ECs relate high-frequency fluctuations for the contemporary period, and for which measurements exist, to slower-changing but important quantities that describe features of future climate change.

2. P.3, l.20-21: This sentence is also not easy to understand, please rephrase.

Similar to the response above, we will also rewrite this sentence in simpler language, noting that if high-frequency changes in the Earth system are ignored, we may be discarding valuable information that can constrain understanding of longer-term climatological variation.

3. P.5, l.10: I wonder if your notation would be simpler if your variable t represented seconds, not years. Then you could absorb the seconds-per-year factor into the frequency ω and drop all the primes for the heat capacity altogether.

   We will certainly consider this. There is always an attraction, of course, to stay in SI units throughout a manuscript.

4. P.5, l.26: There is a "." missing after "Eq".

   We will correct this.

5. P.8, l.22: There is a "." missing after the end of the sentence.

   We will correct this.

6. P.11, l.5: It would be good to add a name for the symbol epsilon here, maybe "error term" or similar.

   We will remind the reviewer at this sentence, in words, that this is a "noise term".

7. P.11, l.16-17: Something is wrong with this sentence.

   We will split this sentence in to two parts, as it carries two messages. Teleconnections can either be constrained by (1) a knowledge of advective winds, or (2) by the differences between two quantities in different locations.

8. P.14, l.17: This reference points to a preprint, please update with the published reference.

   Apologies, we will give the full reference for the Nijsse and Dijkstra paper.

9. Caption of Fig. 1: I think there is a word missing after "This response contains a seasonal (x axis) and long–term (y axis, with seasonality ignored)".

   Yes, the word missing is "variation". We will correct for this.

10. Caption of Fig. 2: "seasonal" forcing instead of "season" forcing. Second to last line: the "measured" value of $\Delta T_S$.

    Thank you – we will correct both of these typos and with the words suggested.

11. Fig. 2: The argument in the cosine of the response term has a different sign than eq. (10). This does not matter due to the symmetry of the cosine, but should be identical to have a consistent notation.

    Of the two choices, we will change the sign of Eqn (10). Using a plus sign feels more natural for increasing time.

12. Fig. 2: The square root in the denominator of the second part of the response is missing. Same for the x and y axis label in (b).

    This is very unfortunate. We created the .pdfs for the diagrams in python and checked the figures carefully after running our script. I had naively assumed that once a .pdf is built, it is the same on all platforms. Unfortunately, the ESD online submission system removed key

characters and symbols from the figures (e.g. the explanatory underbraces of equations terms and related text). The correct diagrams are shown below, and this also answers reviewer points 13 and 14.

13. Figs. 1 and 2: The index "p" is missing for the heat capacity. In addition, sometimes the prime is missing.

    Please see the correct diagrams, presented on the two pages below.

14. Fig. 1 and 2: Why are some parts of the formulas underlined?

    Please see the correct diagrams, presented on the two pages below.

**(a)**

[Figure]

Forcing :
$$H(t) = \underbrace{H_0}_{\substack{\text{Background} \\ \text{Forcing}}} + \underbrace{b\cos(\omega t)}_{\substack{\text{Seasonal} \\ \text{Forcing}}}$$

Model :
$$c'_{p_i} \frac{dT}{dt} = H$$

Response :
$$T_i(t) = \underbrace{\frac{H_0 t}{c'_{p_i}}}_{\substack{\text{Background} \\ \text{Warming}}} + \underbrace{\frac{b}{c'_{p_i}\omega}\sin(\omega t)}_{\substack{\text{Seasonal} \\ \text{Variation}}}$$

**(b)**
$$\frac{\overline{dT_i}}{dt} = \frac{H_0}{c'_{p_i}}$$

[Figure]

$$\Delta T_{s_i} = \frac{2b}{\omega c'_{p_i}}$$

**Figure 1. Schematic representation of a simple emergent constraint.** Panel (a) (top row) shows the combined equation for long–term and seasonal forcing (so with $\omega = 2\pi \, \text{yr}^{-1}$) driving the thermal box model given by Eq. (2) (middle row), and the related response to both forcings, which combine additively to give Eq. (5) (bottom row). Panel (b) illustrates a related emergent constraint, based on the response Eq. (5), as also shown in panel (a). This response contains a seasonal ($x$ axis) and long–term ($y$ axis, with seasonality ignored), and the EC links the two. The EC allows the observation of seasonal fluctuations to constrain the long–term rate of change of state variable, $T$. Each model (black dots, indexed by $i$) has a different implicit value for $c'_p$ i.e. $c'_{p_i}$. The EC is assumed to not be exact, with noise causing variation around the regression line (the $\epsilon_i$ and $\eta_i$ terms of Eq. (1)). The vertical yellow band represents uncertainty in the measurement, $\Delta T_S^*$. The constrained projection of the long–term warming rate (based on the EC, the value of $\Delta T_S^*$ and its uncertainty) is given by the green horizontal band.

**(a)**

**Forcing :**
$$H(t) = -\kappa_i \left.\frac{\partial T_i}{\partial x}\right|_{x=0} = \underbrace{H_0}_{\substack{\text{Background}\\\text{Forcing}}} + \underbrace{b\cos(\omega t)}_{\substack{\text{Seasonal}\\\text{Forcing}}}$$

[Figure]

**Model :**
$$c'_{p_i}\frac{\partial T}{\partial t} = \kappa_i\frac{\partial^2 T_i}{\partial x^2}$$
$$x = 0 \qquad\qquad x \to \infty$$

**Response :**
$$T_i(0,t) = 2H_0\underbrace{\sqrt{\frac{t}{c'_{p_i}\kappa_i\pi}}}_{\substack{\text{Background}\\\text{Warming}}} + \underbrace{\frac{b\cos(\omega t - \pi/4)}{\sqrt{c'_{p_i}\kappa_i\omega}}}_{\substack{\text{Seasonal}\\\text{Variation}}}$$

**(b)**
$$\sqrt{t}\left.\frac{\overline{dT_i}}{dt}\right|_{x=0} = \frac{H_0}{\sqrt{c'_{p_i}\kappa_i\pi}}$$

[Figure]

$$\Delta T_{\text{S}_i}\big|_{x=0} = \frac{2b}{\sqrt{c'_{p_i}\kappa_i\omega}}$$

**Figure 2. Schematic representation of an emergent constraint with a spatial component.** The spatial dimension is defined by $x$. Panel (a) (top row) shows the combined equation for long–term and season forcing at $x = 0$, driving the diffusive model given by Eq. (7) (middle row), and the related response at $x = 0$ and $t > 0$ given by Eqs. (10) and (14) (bottom row). The seasonal forcing (so with $\omega = 2\pi\ \text{yr}^{-1}$) is given by Eq. (8) and the long–term forcing to the thermal model given by Eq. (12). These two forcings generate a response in $T$ at $x = 0$ given by Eqs. (10) and (14) respectively, that combine additively and as shown. Panel (b) illustrates the related emergent constraint, based on the response $T_i(0,t)$ shown in panel (a). This response contains a seasonal ($x$-axis) and long–term ($y$ axis, with seasonality ignored) part, and the EC links the two. The EC allows the observation of seasonal fluctuations to constrain the long–term rate of change. Each model (black dots, indexed by $i$) has a different implicit value for $c'_{p_i} \times \kappa_i$. As for the example of Fig. 1, the EC is again assumed to not be exact, with noise causing variation around the regression line. The vertical yellow band represents uncertainty in the measurement of $\Delta T_S$. The constrained projection of the long–term warming rate (multiplied by $\sqrt{t}$, and based on the EC, the value of $\Delta T_S$ and its uncertainty), is given by the green horizontal band.

---

## Author Response (AR1)

UK Centre for Ecology & Hydrology
Maclean Building, Benson Lane
Crowmarsh Gifford, Wallingford
Oxfordshire
OX10 8BB
UK

T: +44 (0)1491 838800

Prof. Rui A. P. Perdigão
Editor
Earth System Dynamics journal

20th February 2023

Dear Prof. Perdigão

Thank you for helping, guiding and advising us concerning our manuscript:

**"Emergent constraints for the climate system as effective parameters of bulk differential equations"**

We are grateful for the substantial help of the reviewers. We also appreciate your description of our paper as "interesting and relevant".

We have responded to all reviewer requests, as set out in our reply letters. Our changes are additionally illustrated in the marked-up paper version, as attached.

Unlike other papers on emergent constraints (ECs), we accept that what we present here is more abstract. However, at some point, it is necessary to ask, "What is an emergent constraint?" and that leads us into the conceptual field of mathematics. Many of the reviewers' comments relate to linking back better such mathematics to EC features for the actual climate system. Hence we trust that our new paper version achieves the correct level of abstraction, and does not go unnecessarily beyond that.

We hope that the reviewers will be satisfied with our responses. Please do not hesitate to contact me if have any questions concerning our paper.

Thank you again for your help

Yours sincerely

**Chris Huntingford (and on behalf of co-authors)**
Email:   chg@ceh.ac.uk

Response to Reviewer One of our paper: "Emergent constraints for the climate system as effective parameters of bulk differential equations"

Review of Huntingford et al 2023:

On one hand, this paper is clear, well-written, and its PDE examples are simple, relevant, and pleasant to work through. On the other hand, I didn't really learn anything from reading this. For example, I can't imagine anyone understanding Cox et al (2018) without having a deep understanding of the notion that the emergent equations governing temperature changes on various timescales are linked via heat capacity. This left me wondering whether the paper is worth publishing. Ultimately, I think the answer is yes because if someone didn't intuitively understand that emergent constraints occur due to links between underlying governing equations, this paper would do a nice job of introducing them to the concept. I doubt the paper will be cited much, but that doesn't mean it shouldn't be published.

> First, we thank the reviewer for their time assessing our manuscript. We are fully aware that this manuscript will only appeal to a subset of those using Emergent Constraints (ECs) to refine the understanding of climate system components. However, we regard that group as important, representing many with a mathematical interest asking: "What underpins the EC method?". With such a high profile of ECs to constrain uncertainty, we hope the paper will trigger new lines of questioning and understanding of the technique.
>
> Please find our responses below, which are indented and in blue font. Where we quote the revised text directly from the new manuscript version, this is in italic font.

Minor comments (Note my convention is P2 L1 = Page 2, Line 1):

1. P2 L1: observationalists would disagree that ESMs form the basis of climate research. I tend to say they're a pillar of climate research.

   > We have adjusted the manuscript to use the word pillar.

2. P2 L6: It's not accurate to say that ESMs are typically forced with historical and scenario GHGs. A lot of time is spent on PI control, abrupt4xCO2, 1%CO2, etc. Minor rewording is needed.

   > We now note the extensive and often long PI control simulations in the CMIP databases, as well as numerical experiments such as abrupt4xCO2. Specifically, we now add to the manuscript: "*CMIP databases also hold simulations with forcings held at pre-industrial levels to test ESM stability and to characterise their representation of natural variability. Furthermore, there exist illustrative idealised ESM experiments, to determine the response to a continuous cumulative 1% per annum increase in atmospheric $CO_2$, and an abrupt jump by a factor of four in $CO_2$ from pre-industrial levels.*"

3. P2 L19: "main possibly simplest answer" is awkward grammar

   > We have split the sentence and reworded as: "*As standard equations are frequently solved in ESMs, a valid question is: ``why are ESM projections often so different"?. The simplest answer is that some processes are still not fully understood and are therefore parameterised differently between ESMs*"

4. P3 L30 – P4 L2: The first sentence here isn't very clear. I think you are saying that Schlund and others found that ECs based on CMIP5 were generally worse when applied to CMIP6. As written, it sounds like any EC, including ECs developed from CMIP6 data, would have wider

bounds. I also found your wording a bit confusing because wider bounds could come from worse correlations between EC predictor and predictand OR from larger spread in the observations used to constrain. I guess the problem must be the former, but it takes the reader some unnecessary thought to get to that conclusion. Following on this, I think the obvious explanation for larger spread in CMIP6 is that the ECs from CMIP5 were overtrained: they are capturing noise rather than real EC signal. I'm confused how this possibility isn't even in your proposed reasons at all.

> We agree that this text was not as well phrased as it should have been. In our revised manuscript, we have replaced lines P3 L30 to P4 L3, with:
>
> "*Schlund et al. (2020) test the robustness of proposed emergent constraints by out-of-sample testing on a different model ensemble. These researchers found that emergent constraints on ECS, which were developed using the CMIP5 ensemble, do not provide useful constraints on ECS in the CMIP6 models. These ECs, therefore, fail to be "confirmed" (Hall et al., 2019); Fasullo et al., (2015) also provide an important discussion on whether it is expected that ECs hold across different generations of ESMs. Recognising the danger of arriving at spurious emergent constraints based on the results of relatively small model ensembles (Caldwell et al., 2010), Williamson et al. (2021) have set the challenge of deriving more robust theory-based emergent constraints. To inform attempts to meet that challenge, here we address the basic, almost philosophical question: "What is an emergent constraint?"*"

5.  P5 L15-16: you introduce T* here but don't use it again except P8 L3. I suggest deleting both T* references. In particular, the wording of the first intro to T* was very confusing (and I think, unnecessary).

> Our intention is that "*" represents measurements. We have reworded this sentence and made the notation clear, now writing "*Here, and elsewhere, the '*' symbol represents a measurement*". Rather than adjust this notation, we have additionally annotated the word "Data" above the vertical uncertainty bars in Figure 1 and Figure 2 to be "Data, $\Delta T_S$*".
>
> On p8, we are now more explicit, writing "..*a corresponding measurement i.e. observation value, $\Delta T_S$*". (Please see the revised diagrams shown at the end of this document).

6.  P5 L26-28: When you say "running mean", I immediately wonder what the averaging period is. I think it would be better to call this statistic the "annual average". Relatedly, the running mean itself isn't a measure of climate change. The time derivative of the running mean is your proxy for climate change. But of course, the annual average isn't special in this regard – the long-term average of the time derivative of the instantaneous T(t) equation would give the same answer because the derivative is a linear operator.

> Thank you - we do need to remove this ambiguity. Hence, based on this reviewer's request, we reword at this location with: "*A second set of temperature-based statistics we can consider are based on changes in annual means. The time derivative of annual averages is a proxy for global warming. Annual averaging, denoted by an overline, is…*"

7.  P6 paragraph starting L8 and P8 paragraph starting L23: I think this discussion can be improved. I think the big point you're trying to make is that while the fact that there exists a predictive relationship between the observable and the future quantity of interest allows you to predict ECS, the slope of that relationship provides interesting information about the physical equations that underpin that relationship. I think you are further pointing out that even though there may be uncertain terms in the equation governing the current-climate variable and in the equation governing future change, those uncertain terms sometimes cancel out when the quantity you're actually interested in is the ratio between predictor and predictand. As it stands, I don't think it

is interesting that uncertainty in either of 2 parameters would give rise to the intermodel spread needed to compute an emergent constraint. I also don't think you adequately explained why $b_i/H_{0i}$ would be constant across models.

There are two reviewer requests here. The first is general. For a standard research paper on ECs, finding a relationship between the observable and future quantity of interest may be sufficient. However, we provide a potential mathematical explanation for why such regressions emerge. We hope our suggestions enables new ways to interpret and understand ECs, which matters given their widespread use in climate science, and often request by other authors for physical interpretation. We now finish the paper with a very short paragraph of:

*"We note that Brient(2020) argue that when multiple ECs exist to predict the same quantity, each should be weighted by the level of physical understanding they offer to elucidate the relationship. It remains important to understand ECs as they offer an elegant potential capability to lower the continuing uncertainty between ESM projections. In this paper, we suggest an interpretation that ECs are revealing parameters of large-scale differential equations that are implicit within the numerical finite differencing upon which ESMs are built."*

The second request is more specific (P6, L8 and P8, L23). We are keen on a conceptual example where the aggregation of internal behaviours instead applies to forcings, rather than internal system parameters – this might be true for many ESM uncertainties. For simplicity, we also want to maintain a single degree-of-freedom. Hence, we imagine uncertainty to be in the forcing, but identical between seasonal and long-term forcing (i.e. this implies ratio $b_i/H_{0i}$ fixed between ESMs, even if the $b_i$ and $H_{0i}$ values vary). We accept that this appears slightly contrived, but it does allow us to illustrate a key point in our conceptual structure.

We have improved the wording at the two points, providing a better logical description. For P6, we now amend to: *"However, an alternative possibility is an EC where there is instead uncertainty in the magnitude of the forcing of an Earth system component (rather than the inter-ESM spread in how the component itself is modelled). For instance, there remains a range of representations between ESMs, of the translation of atmospheric aerosol levels to their cooling effect. Now the forcing parameters are indexed as $b_i$ and $H_{0i}$, although we imagine for each ESM, the uncertainty for each is similar and so the ratio bi/H0i is invariant between models. This setup yields an EC of identical form to that of Fig. 1, but instead cp' has a single numerical value, common to all ESMs."*

Then at P8, we have tightened the text, to now write: *"As for the discussion of uncertainty in the forcing boundary conditions of the box model, and their potential constraint, the same possibility exists for our example with spatial variability. Should effective parameters $c_p$ and $\kappa$ show little or no variation between ESMs, yet there is uncertainty in b of Eq. (8) and $H_0$ of Eq. (12) (and both parameters have similar unknowns, so again $b/H_0$ is invariant inter-ESMs), then the EC combined with data for $\Delta T_S$ acts to remove that forcing uncertainty."*

8. This is a minor point, but the seasonal cycle in eq 4 and eq 11 won't be exactly equal to the observed seasonal cycle in a warming planet ($H_0 > 0$) because the planet will have warmed a bit in the 6 months between winter and summer.

Eqn. (5) introduces the background warming component (via parameter $H_0$), so this small adjustment would be seen when considering any yearly period predicted by that equation. However, we agree the point the reviewer raises is worth stating explicitly. Hence we now write, where the seasonal cycle is introduced, *"..form of known seasonal cycle, unaffected by any background trends, and this forcing results in a model-specific temperature, $T_i(t)$."*

9.  P8 L17: defining your current-climate metric as d/dt(annual-ave T(x=0)) makes sense, but multiplying it by sqrt(t) seems contrived. If you have such an exact understanding of the underlying equations, you'd probably already know what $H_0$ was, so a regression would be unnecessary!

> Process knowledge of any heat conduction (i.e. "parabolic") PDE often suggests that sqrt(t) behaviour is present. Ideally, the "*y*"-axis of any emergent constraint is independent of time (or scenario), as shown. This query does raise an interesting yet rarely mentioned point, that in some instances, an EC may be robust but its position on an '*x-y*' graph depends on a future time or GHG level.
>
> Hence, we add an additional sentence that makes this point, noting that should the '*y*'-axis be simply dT/dt, then the EC would move downwards by 1/sqrt(t). We write: "*As an aside, in the y-axis of Fig 2, we retain the √t factor to make the vertical position of the EC in the diagram independent of time or GHG level.*"

10. P11 L34-P12 L1: I can't imagine how a real emergent constraint wouldn't have a physical underpinning that can be expressed as an equation. We may not know what that equation is, but if there truly isn't an underlying equation behind an empirical relationship, how could that relationship possibly be real?

> The original paper version contains the sentence (P11, L34): "We do not propose this as a universal theory of ECS, as some may function for other reasons". Based on this reviewer's comment, we have removed that sentence. The manuscript will then lead directly into the next sentence that discusses the importance of an underlying physical process. This second sentence is written more strongly as: "*With the prevailing view that physical intuition should guide EC discoveries, rather than e.g. data mining, our suggestion, therefore, supports that standpoint*".

I don't think the emergent relationships that distill into an EC are necessarily (or even typically) PDEs. In your examples, the fine scale governing equations are PDEs, but the equations you derive for seasonal cycle and warming tendency are not. Similarly, concepts like "if you don't have much cloud in the current climate, then you don't have much cloud to lose in the future" are fundamentally connecting model state to model change. This doesn't invalidate any of this work, but a reframing of the title and some rewording in the abs

> Some large-scale implicit equations may be ODEs (as per our first example) or, indeed, simply algebraic. In the other direction, the overall system may still be relatively simple but involves coupled equations. We have modified the Abstract to remove mention of "PDEs" and replace it simply with "*differential equations*". An extra sentence is now added to the Discussion to note that the equations may take many formats. We write: "*The equation forms may be PDEs, they may be coupled, or could be simply ordinary differential equations or in algebraic form*"
>
> Revised Figure 1:

**(a)**

Forcing :

$$H(t) = \underbrace{H_0}_{\substack{\text{Background}\\\text{Forcing}}} + \underbrace{b\cos(\omega t)}_{\substack{\text{Seasonal}\\\text{Forcing}}}$$

Model :

$$c'_{p_i}\frac{\mathrm{d}T}{\mathrm{d}t} = H$$

Response :

$$T_i(t) = \underbrace{\frac{H_0 t}{c'_{p_i}}}_{\substack{\text{Background}\\\text{Warming}}} + \underbrace{\frac{b}{c'_{p_i}\omega}\sin(\omega t)}_{\substack{\text{Seasonal}\\\text{Variation}}}$$

**(b)**

[Figure]

$$\overline{\frac{\mathrm{d}T_i}{\mathrm{d}t}} = \frac{H_0}{c'_{p_i}}$$

Data, $\Delta T_S^*$

Emergent Constraint

Constrained Projection

Annual running mean warming rate (K yr$^{-1}$)

Decreasing $c'_{p_i}$

Seasonal range (K)

$$\Delta T_{S_i} = \frac{2b}{\omega c'_{p_i}}$$

Revised Figure 2:

**(a)**

Forcing :
$$H(t) = -\kappa_i \left.\frac{\partial T_i}{\partial x}\right|_{x=0} = \underbrace{H_0}_{\substack{\text{Background}\\\text{Forcing}}} + \underbrace{b\cos(\omega t)}_{\substack{\text{Seasonal}\\\text{Forcing}}}$$

Model :
$$c'_{p_i}\frac{\partial T}{\partial t} = \kappa_i\frac{\partial^2 T_i}{\partial x^2}$$

$x = 0 \qquad\qquad x \to \infty$

Response :
$$T_i(0,t) = \underbrace{2H_0\sqrt{\frac{t}{c'_{p_i}\kappa_i\pi}}}_{\substack{\text{Background}\\\text{Warming}}} + \underbrace{\frac{b\cos(\omega t - \pi/4)}{\sqrt{c'_{p_i}\kappa_i\omega}}}_{\substack{\text{Seasonal}\\\text{Variation}}}$$

[Figure]

**(b)**
$$\sqrt{t}\,\left.\overline{\frac{\mathrm{d}T_i}{\mathrm{d}t}}\right|_{x=0} = \frac{H_0}{\sqrt{c'_{p_i}\kappa_i\pi}}$$

[Figure]

Data, $\Delta T_S^*$

Emergent Constraint

Constrained Projection

Annual running mean warming rate $\times\sqrt{t}$ (K yr$^{-1/2}$)

Decreasing $c'_{p_i}\kappa_i$

Seasonal range (K)

$$\left.\Delta T_{\mathbf{S}_i}\right|_{x=0} = \frac{2b}{\sqrt{c'_{p_i}\kappa_i\omega}}$$

Response to Reviewer Two of our paper: "Emergent constraints for the climate system as effective parameters of bulk differential equations"

Review of Huntingford et al 2023:

The paper "Emergent constraints for the climate system as effective parameters of bulk differential equations" by Chris Huntingford et al. provides a formal description of emergent constraints as parameters of large-scale partial differential equations (PDEs). In contrast to small-scale PDEs explicitly coded into Earth system models (ESMs), these large-scale PDEs are not directly included in the models, but emerge across ESMs when aggregated across larger scales. Huntingford et al. provide two example PDEs derived from simple thermal models. By assuming different bulk parameters (e.g., heat capacities) for the different ESMs, they show that these PDEs can be used to derive emergent relationships between short-term and long-term responses of the system, which ultimately can be used as emergent constraints with appropriate measurements of the real Earth system.

> First, we thank the reviewer for their time assessing our manuscript.
>
> We appreciate the reviewer's summary above. As noted, our view of many ECs is that the "emergent" property is the discovery of large-scale differential equations coded implicitly in ESMs (via aggregation of explicit coding at finer scales). To ensure this is clear, we have amended the Abstract sentence at line 17 to read: "*We suggest that many ECs link to effective hidden differential equations implicit in ESMs and which aggregate small-scale features*"

**General Comments**

This paper reads well and provides an interesting approach that allows the derivation of emergent constraints from bulk PDEs. I agree with the authors that an emergent constraint discovery method based on physical reasoning and mathematical models is much more desirable than data mining, and will eventually lead to more credible and robust emergent constraints. However, I have some concerns about the relevance of this study regarding "real" emergent constraints.

> We are grateful that the reviewer thinks our paper reads well and is an interesting approach. We take full account of their concerns listed below, responding in full. Our replies are in blue font and indented, and use an italic font where we quote new text from the manuscript.
>
> Concerning some of the more technical points, please note that there was an issue with the diagram .pdfs and the ESD online converter. The correct diagrams are presented below, and if our manuscript is accepted, we will work carefully with ESD to make sure they are reproduced as expected.

Currently, a large part of the argumentation of the paper is based on two very simple PDEs. Especially in the context of a changing climate (which is a necessary condition here), I think the equations are too simplified. Since the PDEs are missing a "loss" term, a constant forcing will lead to an infinitely rising temperature, which is not realistic. For example, what happens if you add linear loss terms (linear feedback) $-\lambda*T$ to your PDEs (e.g., so that your eq. (2) is similar to eq. (1) of Cox et al. 2018)? Could you still derive the emergent relationships from these new equations? I can imagine that there are certain conditions (e.g., small times, small $\lambda$, large forcings, …) under which your original equations are good approximations, but it would be good to guide the reader in detail through this process. Additionally, it would be very helpful if you can provide more details on these emerging bulk equations themselves and why they should be present in an ensemble of ESMs. Do you have any recommendations how to find such PDEs? An example with a real emergent constraint would also be incredibly helpful. All this will ultimately help the reader to gain more trust in your framework.

> The reviewer asks some fascinating questions here but answering these is cutting-edge research that is beyond the scope of this initial short perspective paper. However, we are keen to acknowledge the points made, and we add text to acknowledge the nature of the challenges that the reviewer poses. Specifically, we now write in the Discussion:
>
> "*To aid transparency, we have also assumed underlying PDEs that are simple by design. Making these underlying models more relevant to the Earth's climate is an outstanding challenge. For example, in addition to horizontal heat transport, our planet emits longwave radiation to the wider universe. Such radiation provides the restoring force, $\lambda$, that ultimately stabilises the near-surface temperature. Including such a restoring force in our simple PDE models is one possible extension of our analysis, although, in tandem with an unknown heat capacity, $c_p$, this would potentially generate a two-dimensional EC. In practice, fitting a two-dimensional EC may be challenging given the relatively small number of data points (i.e. individual ESMs). Furthermore, analytical solutions may exist that allow for a time-varying value of H that approximates known historical climatic forcing*".*

Finally, two technical comments: first, it would be very helpful if you could use continuous line numbers (and not start with "1" on every page) and also add line numbers to figure captions. Second, please consider depositing your code in a publicly accessible repository (e.g., Zenodo) to make your analysis more transparent and reproducible for other researchers.

> Unfortunately, I think the ESD template for submission causes this form of line numbering. Final ESD papers have no line numbers.
>
> We are very happy to upload our code to a standard scientific repository. We have registered with GitHub and then linked to Zenodo, which has given us a fixed doi for this paper of:
>
> https://doi.org/10.5281/zenodo.7633839

**Specific Comments**

1. P.2, l.30: Maybe add a reference here? E.g., Knutti et al. (2017), https://doi.org/10.1002/2016GL072012

   > Thank you. We have added this reference, and at the location suggested.

2. P.3, l.4: It would be more precise to refer to "observational" data here (alternatively "observation-based").

   > We will make this suggested wording alternation.

3. P.3, l.12: A better reference for this might be Hall & Qu (2006), https://doi.org/10.1029/2005GL025127. You might also want to cite Allen & Ingram (2002), https://doi.org/10.1038/nature01092 here.

   > Yes, these two references are more appropriate. We have adjusted the manuscript accordingly.

4. P.4, l.1-2: It might be helpful for the reader to add the key conclusion(s) of the discussion of Fasullo et al. (2015) you mention here.

   > We now write: "*Fasullo et al. (2015) also provide a key discussion on whether it is expected that ECs hold across different generations of ESMs. Those authors argue that additional*

*processes identified as important but uncertain, and introduced to newer ensembles, could generate ECs that make different predictions. Fasullo et al. (2015) provide the example of models characterising convection, and its impact on simulated cloud features, which ultimately may alter EC estimates of ECS."*

5.  P.4, l.29: I guess technically it's a function of the total noise, so $\varepsilon$ **and** $\eta$, not only $\varepsilon$.

    Correct. That is amended to read $\varepsilon+\eta$.

6.  P.5, l.18: Required for what?

    We will rewrite this as: "*ECs require a quantity that is both modelled for the contemporary period and is available as a measurement, such as the seasonal range, $\Delta T_S$.*

7.  P.6, l.15: It's not only the data points (I guess by "data points" you are referring to the (x, y) tuples you get from the models?), but also the measurements that constrains the forcing element b.

    Please see our response directly below, which concerns the same sentence.

8.  P.6, l.15-16: I think this sentence is not clear enough: "With the forcing uncertainties common for both short– and long–term drivers". You need to explicitly assume that $b_i/H_{0i}$=const across models; you should mention that.

    We have rewritten this sentence (and split it into two), as well as adjusting the sentence that follows it. We now write: "*In this case, the emergent constraint represents the discovery that there is a single ESM-independent internal bulk parameter (i.e. $c_p'$). Measurements then provide the constraint to remove uncertainty in the forcing element $b_i$. With the forcing uncertainties common for both short-and long-term drivers (i.e. the assumption that $b_i/H_{0i}$ is constant), the measurements implicitly constrain $H_{0i}$, and thus the background warming, dT/dt.*"

9.  P.6, eq. (8): You might want to refer to Fourier's law here.

    Just before Eq. (8), we now write: "*We ...start by prescribing a seasonal boundary condition (Fourier's law of heat conduction),...*"

10. P.8, l.17: Why don't you simply divide T(0, t) by sqrt(t) to get a y that is not dependent on t?

    Yes, we do exactly that scaling as the '$y$'-axis of the EC (please see Figure 2). Our other reviewer requested that we consider the opposite, of not normalising by sqrt(t). We hope the current framework of keeping in the sqrt(t) in the text but normalising in the EC diagram (so making the EC time-independent) is a satisfactory presentation.

    We have added to the paper the following sentence: "*As an aside, in the y-axis of Fig 2, we retain the √t factor to make the vertical position of the EC in the diagram independent of time or GHG level.*"

11. P.12, l.10: I think this classification only applies to linear second-order PDEs, not to every PDE.

    Thank you – we now write "*for instance, every second-order PDE being either diffusive, elliptic or parabolic*".

12. P.12, l.10-12: Can you elaborate what you exactly mean by these "one-to-one mappings" and why this should be the case? This is not clear to me.

We have rewritten this in more straightforward language. We now write: "*We suggest that the perspective offered here may open ways to classify ECs based on the type of any discovered underpinning equations they link to. Confirming such links may also allow the study of some aspects of climate change from a more analytical applied mathematics standpoint.*"

**Technical Corrections**

1. P.3, l.19-20: The second part of this sentence is hard to understand, please rephrase.

We have rewritten this sentence more simply and clearly (including splitting the sentence). We state that many ECs relate high-frequency fluctuations for the contemporary period, and for which measurements exist, to slower-changing and important quantities that describe features of future climate change. We now write: "*Notable is that for many discovered ECs, the modelled quantity that is also measured during the contemporary period is often a high-frequency statistic or attribute of the climate system. The EC relates this quantity that fluctuates at shorter timescales to a longer-term attribute of the Earth system relevant to projecting how climate will respond to rising GHG concentrations.*"

2. P.3, l.20-21: This sentence is also not easy to understand, please rephrase.

We rephrase this sentence to make it clearer that if high-frequency changes in the Earth system are ignored, we may be discarding valuable information that can constrain understanding of longer-term climatological variation. We now write: "*The ability of ECs to use knowledge of contemporary high-frequency variations to constrain understanding of expected future climate change highlights how ignoring fluctuations at short timescales may constitute disregarding valuable information*".

3. P.5, l.10: I wonder if your notation would be simpler if your variable t represented seconds, not years. Then you could absorb the seconds-per-year factor into the frequency ω and drop all the primes for the heat capacity altogether.

In this one instance, we respectfully request we leave the paper "as is", and the units as years. While it is often preferable to retain SI units (i.e. seconds) in papers describing specific processes, for this more perspective-orientated paper, the units of years are more intuitive.

4. P.5, l.26: There is a "." missing after "Eq".

We have corrected this.

5. P.8, l.22: There is a "." missing after the end of the sentence.

We have corrected this.

6. P.11, l.5: It would be good to add a name for the symbol epsilon here, maybe "error term" or similar.

We have reminded the reader of this by adding just before the ε term the words "*noise term*".

7. P.11, l.16-17: Something is wrong with this sentence.

The original version of this sentence was poorly worded. We now split this sentence into two parts, as it carries two messages. Teleconnections can either be constrained by (1) knowledge of advective winds, or (2) by the differences between two quantities in different locations.

We now write: "*For instance, in addition to our example of diffusion, ECs may reveal implicit PDEs with an advective component that corresponds to atmospheric transport. In many cases, atmospheric transport provides the coupling between two spatially-distant components of the Earth system, generating what is often called a ``teleconnection''. To constrain the strength of future teleconnections, an EC is likely to need a present-day measurement of ....*"

8.  P.14, l.17: This reference points to a preprint, please update with the published reference.

Apologies, we now give the full published reference for the Nijsse and Dijkstra paper.

9.  Caption of Fig. 1: I think there is a word missing after "This response contains a seasonal (x axis) and long–term (y axis, with seasonality ignored)".

Yes, the word missing is "*variation*". We have corrected this.

10. Caption of Fig. 2: "seasonal" forcing instead of "season" forcing. Second to last line: the "measured" value of $\Delta T_S$.

Thank you – we have corrected both of these typos with the words suggested.

11. Fig. 2: The argument in the cosine of the response term has a different sign than eq. (10). This does not matter due to the symmetry of the cosine, but should be identical to have a consistent notation.

Of the two choices, we have changed the sign (within the "cos") of Eqn (10). Using a plus sign feels more natural for increasing time.

12. Fig. 2: The square root in the denominator of the second part of the response is missing. Same for the x and y axis label in (b).

This is very unfortunate. We created the .pdfs for the diagrams in python (matplotlib calling embedded latex for annotation) and checked the figures carefully after running our script. I had naively assumed that once a .pdf is built, it is the same on all platforms. Unfortunately, the ESD online submission system removed key characters and symbols from the figures (e.g. the explanatory underbraces of equations terms and related text).

After much consideration, to avoid any risk of mistranslation of embedded latex fonts, the figures are now saved in .png format. While this is a picture rather than vector format (and so it losses a little sharpness under high levels of zooming in), this lowers the risk of a reader not seeing the intended diagram correctly.

The correct diagrams, with captions, are shown below - this also answers reviewer points 13 and 14.

13. Figs. 1 and 2: The index "p" is missing for the heat capacity. In addition, sometimes the prime is missing.

Please see the correct diagrams, presented on the two pages below.

14. Fig. 1 and 2: Why are some parts of the formulas underlined?

Please see the correct diagrams, presented on the two pages below.

**(a)**

Forcing : $H(t) = \underbrace{H_0}_{\substack{\text{Background} \\ \text{Forcing}}} + \underbrace{b\cos(\omega t)}_{\substack{\text{Seasonal} \\ \text{Forcing}}}$

Model : $c'_{p_i}\dfrac{\mathrm{d}T}{\mathrm{d}t} = H$

Response : $T_i(t) = \underbrace{\dfrac{H_0 t}{c'_{p_i}}}_{\substack{\text{Background} \\ \text{Warming}}} + \underbrace{\dfrac{b}{c'_{p_i}\omega}\sin(\omega t)}_{\substack{\text{Seasonal} \\ \text{Variation}}}$

**(b)**

[Figure]

$\dfrac{\overline{\mathrm{d}T_i}}{\mathrm{d}t} = \dfrac{H_0}{c'_{p_i}}$

Data, $\Delta T_S^*$

Emergent Constraint

Constrained Projection

Annual running mean warming rate (K yr$^{-1}$)

Decreasing $c'_{p_i}$

Seasonal range (K)

$\Delta T_{S_i} = \dfrac{2b}{\omega c'_{p_i}}$

**Figure 1. Schematic representation of a simple emergent constraint.** Panel (a) (top row) shows the combined equation for long–term and seasonal forcing (so with $\omega = 2\pi$ yr$^{-1}$) driving the thermal box model given by Eq. (2) (middle row), and the related response to both forcings, which combine additively to give Eq. (5) (bottom row). Panel (b) illustrates a related emergent constraint, based on the response Eq. (5), as also shown in panel (a). This response contains a seasonal ($x$ axis) and long–term ($y$ axis, with seasonality ignored) variation, and the EC links the two. The EC allows the observation of seasonal fluctuations to constrain the long–term rate of change of state variable, $T$. Each model (black dots, indexed by $i$) has a different implicit value for $c'_p$ i.e. $c'_{p_i}$. The EC is assumed to not be exact, with noise causing variation around the regression line (the $\epsilon_i$ and $\eta_i$ terms of Eq. (1)). The vertical yellow band represents uncertainty in the measurement, $\Delta T_S^*$. The constrained projection of the long–term warming rate (based on the EC, the value of $\Delta T_S^*$ and its uncertainty) is given by the green horizontal band.

[Figure]

**(a)**
Forcing : $H(t) = -\kappa_i \left.\dfrac{\partial T_i}{\partial x}\right|_{x=0} = \underbrace{H_0}_{\substack{\text{Background} \\ \text{Forcing}}} + \underbrace{b\cos(\omega t)}_{\substack{\text{Seasonal} \\ \text{Forcing}}}$

Model : $\quad c'_{p_i} \dfrac{\partial T}{\partial t} = \kappa_i \dfrac{\partial^2 T_i}{\partial x^2}$

$x = 0 \hspace{5cm} x \to \infty$

Response : $\quad T_i(0,t) = \underbrace{2H_0 \sqrt{\dfrac{t}{c'_{p_i}\kappa_i\pi}}}_{\substack{\text{Background} \\ \text{Warming}}} + \underbrace{\dfrac{b\cos(\omega t - \pi/4)}{\sqrt{c'_{p_i}\kappa_i\omega}}}_{\substack{\text{Seasonal} \\ \text{Variation}}}$

**(b)** $\sqrt{t}\left.\overline{\dfrac{dT_i}{dt}}\right|_{x=0} = \dfrac{H_0}{\sqrt{c'_{p_i}\kappa_i\pi}}$

Data, $\Delta T_S^*$ — Emergent Constraint

Constrained Projection

Decreasing $c'_{p_i}\kappa_i$

Annual running mean warming rate $\times\sqrt{t}$ (K yr$^{-1/2}$)

Seasonal range (K) $\qquad \left.\Delta T_S\right|_{x=0} = \dfrac{2b}{\sqrt{c'_{p_i}\kappa_i\omega}}$

**Figure 2. Schematic representation of an emergent constraint with a spatial component.** The spatial dimension is defined by $x$. Panel (a) (top row) shows the combined equation for long–term and seasonal forcing at $x = 0$, driving the diffusive model given by Eq. (7) (middle row), and the related response at $x = 0$ and $t > 0$ given by Eqs. (10) and (14) (bottom row). The seasonal forcing (so with $\omega = 2\pi$ yr$^{-1}$) is given by Eq. (8) and the long–term forcing to the thermal model given by Eq. (12). These two forcings generate a response in $T$ at $x = 0$ given by Eqs. (10) and (14) respectively, that combine additively and as shown. Panel (b) illustrates the related emergent constraint, based on the response $T_i(0,t)$ shown in panel (a). This response contains a seasonal ($x$-axis) and long–term ($y$ axis, with seasonality ignored) part, and the EC links the two. The EC allows the observation of seasonal fluctuations to constrain the long–term rate of change. Each model (black dots, indexed by $i$) has a different implicit value for $c'_{p_i} \times \kappa_i$. As for the example of Fig. 1, the EC is again assumed to not be exact, with noise causing variation around the regression line. The vertical yellow band represents uncertainty in the measured value of $\Delta T_S$. The constrained projection of the long–term warming rate (multiplied by $\sqrt{t}$, and based on the EC, the value of $\Delta T_S$ and its uncertainty), is given by the green horizontal band.

---

## Author Response (AR2)

UK Centre for Ecology & Hydrology
Maclean Building, Benson Lane
Crowmarsh Gifford, Wallingford
Oxfordshire
OX10 8BB
UK

T: +44 (0)1491 838800

Prof. Rui A. P. Perdigão
Editor
Earth System Dynamics journal

23rd March 2023

Dear Prof. Perdigão

Thank you for the additional help with our paper:

**"Emergent constraints for the climate system as effective parameters of bulk differential equations"**

We have implemented the requested edits and typos spotted by the reviewer.

We appreciate the editorial view that this manuscript has the potential to provide much underpinning theoretical understanding of the technique of emergent constraints.

The latest LaTeX files are now uploaded to the ESD server. Please do not hesitate to contact me if there are any further queries.

Thank you again for your help.

Yours sincerely

**Chris Huntingford (and on behalf of co-authors)**
Email: chg@ceh.ac.uk